# Dynamic switching from coherent perfect absorption to parametric amplification in a nonlinear spoof plasmonic waveguide

Wen Yi Cui [1,2], Jingjing Zhang [1,2] ✉, Yu Luo [3] ✉, Xinxin Gao[4] & Tie Jun Cui [1,2] ✉

Coherent perfect absorption (CPA) and amplification of electromagnetic waves are converse phenomena, where incoming radiations are coherently dissipated or amplified by structured incidences. Realizing such two phenomena simultaneously in a single device may benefit various applications such as biological sensing, photo detection, radar stealth, solar-thermal energy sharing, and wireless communications. However, previous experimental realizations of CPA and amplification generally require precise controls to the loss and gain of a system, making dynamic switching between the absorption and amplification states a challenge. To this end, we propose a nonlinear approach to realize CPA and parametric amplification (PA) simultaneously at the same frequency and demonstrate experimentally dynamic switch from the CPA to PA states in a judiciously designed nonlinear spoof plasmonic waveguide. The measured output signal gain can be continuously tuned from −33 dB to 22 dB in a propagation length of 9.2 wavelengths. Compared to the traditional linear CPA, our approach relaxes the stringent requirements on device dimensions and material losses, opening a new route to actively modulate the electromagnetic waves with giant amplification-to-absorption contrast in a compact platform. The proposed nonlinear plasmonic platform has potential applications in on-chip systems and wireless communications.

In microwave and optical engineering, absorption[1-11] and amplification[12-22] are widely used in signal processing and wireless communication systems such as telephone, radio, television, and radar to adjust the signal amplitude to the appropriate values, or to achieve sensing and photodetection. In general, the coherent amplification can be achieved by incorporating gain materials to amplify a specific optical mode. In contrast, coherent perfect absorption (CPA), which arises from time-reversed process of lasing[23], can be achieved through the destructive interference between the reflected and transmitted waves. Coherent perfect absorbers are especially useful in applications which require the absorption of a coherent optical mode at a fixed frequency[24-33]. With the rapid development of coherent perfect absorbers, recent works have achieved multi-channel[34-36], broadband[37,38], or arbitrary wavefront absorption[39] capabilities. In these applications, it is also desirable to realize absorption and amplification of the same optical mode simultaneously with high absorption rate and high gain. In the past, multi-functional systems capable of coherent perfect absorption, perfect transmission, and small-signal amplification were proposed and explored theoretically[31,40–43], and only a few of them were

[1]State Key Laboratory of Millimeter Waves, Southeast University, Nanjing 210096, China. [2]Institute of Electromagnetic Space, Southeast University, Nanjing 210096, China. [3]National Key Laboratory of Microwave Photonics, Nanjing University of Aeronautics and Astronautics, Nanjing 211106, China. [4]State Key Laboratory of Terahertz and Millimeter Waves, City University of Hong Kong, Hong Kong SAR, China. ✉e-mail: zhangjingjing@seu.edu.cn; yu.luo@nuaa.edu.cn; tjcui@seu.edu.cn

demonstrated by experiments (e.g., lasing and anti-lasing on the same structure[44]).

However, most of previous studies are still based on linear approaches where the realization of CPAs requires precise control of the system parameters, including the cavity size and gain/loss of the material. Up to day, it is still challenging to dynamically tune the system from absorption to amplification with a large modulation depth. For example, Fig. 1a illustrates a simple type of tunable coherent perfect absorber similar to the Dallenbach absorber[45-48], consisting of a gain-loss medium layer on top of a metallic ground plane. When an input wave is incident upon a lossy medium, part of the wave is reflected at the first interface, and the rest is transmitted. The transmitted wave is then reflected by the second interface and subsequently interferes with the first reflected wave. When the destructive interference occurs, the reflection is reduced, and the absorption is enhanced. CPA requires zero reflection at the working frequency, namely $\tan k_1 L = i\sqrt{\varepsilon_1}$, in which $\varepsilon_1$, $k_1$ and $L$ are the relative permittivity, wave number and length of the gain-loss material, respectively. For a given frequency, this condition can be fulfilled only at some specific slab thicknesses and with a fixed loss tangent of the absorbing material, as shown in Fig. 1b. Switching from CPA to the amplification state requires tuning the loss tangent from positive to negative. Unfortunately, materials whose loss tangent is tunable over such a wide range

are only available at limited frequencies. Hence, a general method which can be applied to any frequency range is still highly desired.

To tackle such a challenge, we propose an ultrathin nonlinear spoof surface plasmon polariton (SSPP) structure that can realize dynamic switching from CPA to parametric amplification (PA), by modulating only the phase difference between the input waves. The proposed nonlinear system has several distinct advantages, (1) it is insensitive to the material's loss tangent and the CPA/PA effect can be achieved within a certain range of device dimensions; (2) the phase of the input wave can be easily tuned by a phase shifter, making our design easy to implement at any frequency. A theory is developed to describe the loss/gain properties of the proposed nonlinear SSPP system. Through both theoretical calculations and experimental measurements, we demonstrate that the power of the transmitted signal wave can be continuously tuned from 22 dB gain to −33 dB attenuation.

## Results

### Theory of nonlinear CPA and parametric amplification

Figure 1c illustrates the conceptual basis of the nonlinear CPA proposed in this work. A large pump wave and a small signal wave are coupled together into a nonlinear slab with a thickness $L$. The frequency of the differential frequency wave generated from the three-

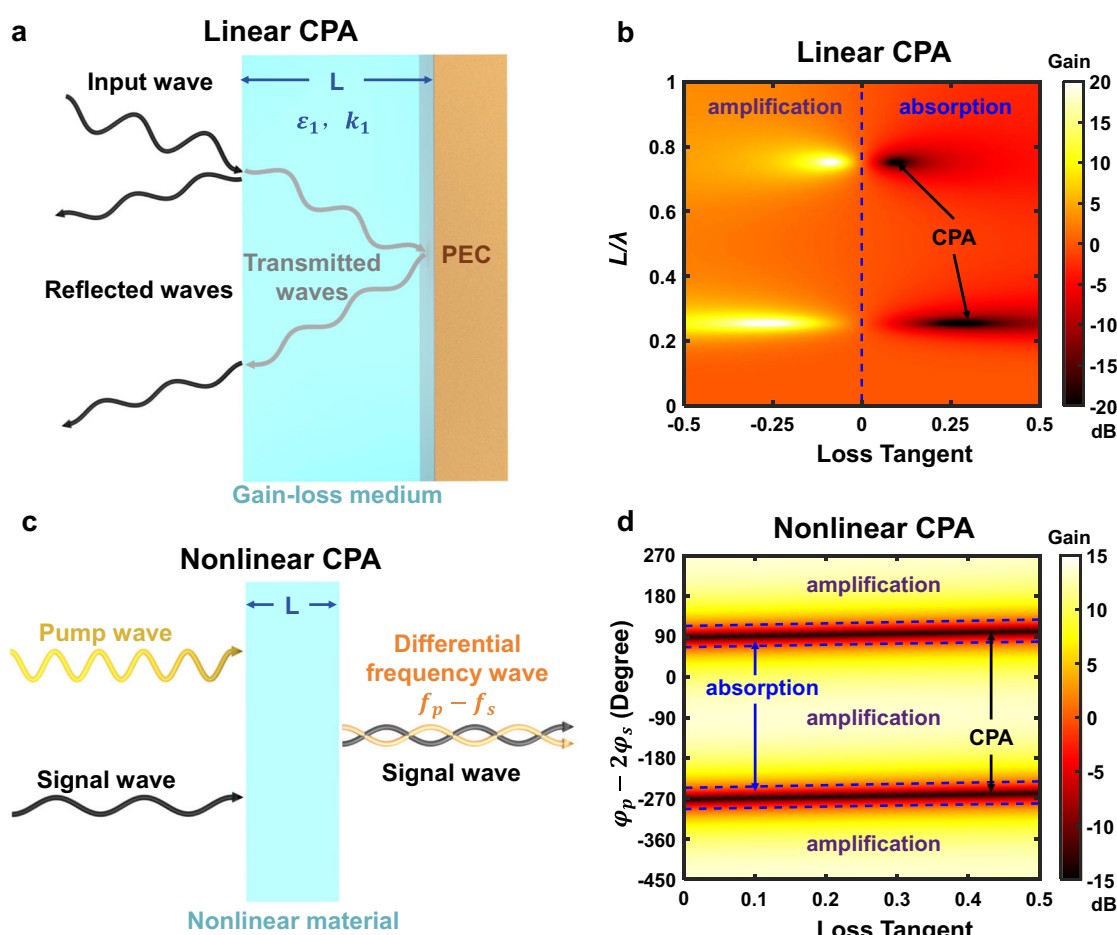

**Fig. 1 | Principles of linear and nonlinear coherent perfect absorptions (CPAs).** **a** Schematic of a linear coherent perfect absorber consisting of a gain-loss medium layer on top of a metallic ground plane, in which PEC indicates perfectly electric conductor. The destructive interference between the waves reflected at the first and second interfaces results in the enhancement of absorption. **b** Gain/absorption of the linear system in terms of the thickness $L$ and loss tangent of the gain-loss medium layer. CPA can be realized only at some specific slab thicknesses and with a

fixed loss tangent of the absorbing material. **c** Schematic of the nonlinear CPA. By controlling the phase difference between the input signal and pump waves, the interference between the transmitted signal and generated differential frequency waves from a nonlinear medium can be tuned from constructive to destructive to achieve PA and CPA. **d** Gain/absorption of the signal wave in terms of the phase difference $\varphi_p - 2\varphi_s$ and the loss tangent of the nonlinear medium.

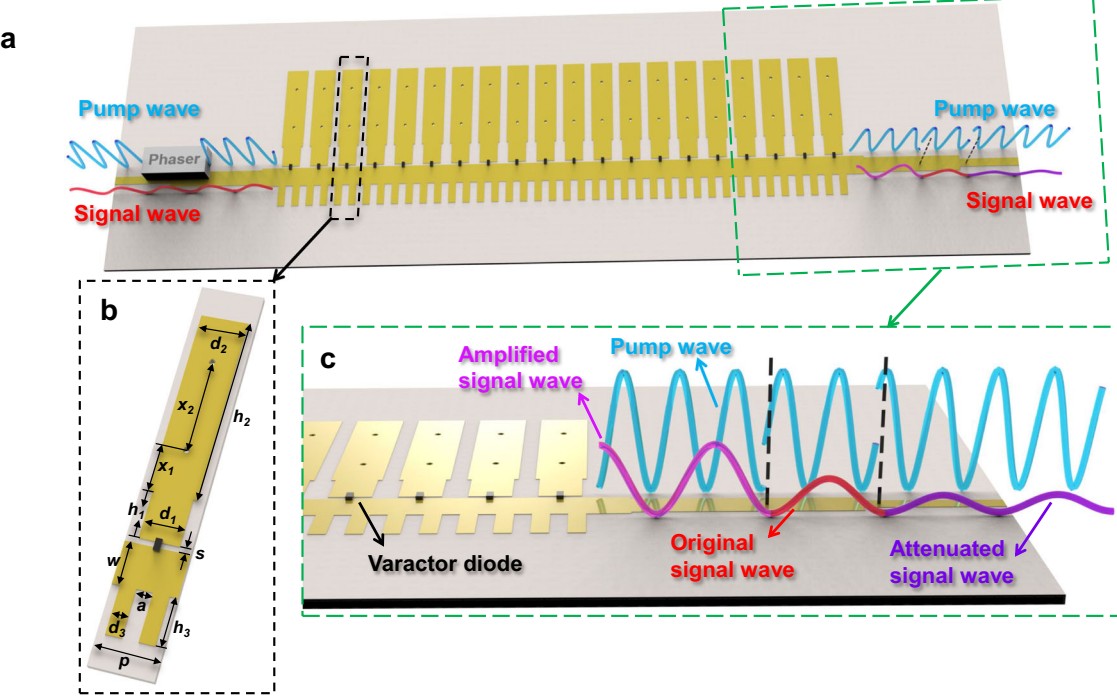

**Fig. 2 | The nonlinear spoof surface plasmon polariton (SSPP) waveguide for dynamic switching of signal waves from coherent perfect absorption to parametric amplification. a** A schematic of the nonlinear SSPP waveguide. **b** The unit structure of the SSPP waveguide. **c** Details of the output port.

wave mixing process is the same as that of the signal wave, indicating that the differential frequency wave and the signal wave can interfere coherently. By controlling the phase difference between the input signal and pump waves, the interference between the transmitted signal and generated differential frequency waves can be tuned from constructive to destructive to achieve PA and CPA, respectively. If the frequencies of the pump wave and signal wave are exchanged, the second harmonic of the pump wave can be used to eliminate the signal wave, but not to amplify it (detailed discussions are shown in Supplementary Note 5).

To give a theoretical description, we assume that the waves propagate along the $z$ direction, where, the electric fields of the signal and pump waves take the form of: $E_s(z) = A_s(z)e^{ik_s z - i\omega_s t}$ and $E_p(z) = A_p(z)e^{ik_p z - i\omega_p t}$, where $A_s$, $A_p$, $\omega_s$ and $\omega_p$ are respectively the electric field amplitudes and angular frequencies of the signal and pump waves. Since the equivalent resistance of the varactor diodes we used in the nonlinear SSPP unit cannot be ignored, both signal and pump waves are depleted along the propagation. Thus, the wave number $k$ has the real part $k'$ (representing the propagation factor) and the imaginary part $k''$ (representing the transmission loss), i.e., $k_s = k'_s + ik''_s$, and $k_p = k'_p + ik''_p$. The amplitude of the pump wave $A_p(z)$ is a complex constant $A_p = |A_p|e^{i\varphi_p}$; $\varphi_s$ and $\varphi_p$ are respectively the initial phases of the incident signal and pump waves.

When $f_p = 2f_s$ and $k'_p = 2k'_s$, the nonlinear coupling process between the signal and pump waves can be described by the following partial differential equation,

$$\frac{d^2 A_s(z)}{dz^2} + k''_p \frac{dA_s(z)}{dz} - \frac{\omega_s^4 |\chi_{eff}^{(2)}|^2 |A_p|^2}{|k_s|^2 c_0^4} e^{-2k''_p z} A_s(z) = 0, \quad (1)$$

where $\chi_{eff}^{(2)}$ is the second-order effective nonlinear coefficient. The general solution of above equation is obtained as

$$A_s(z) = A_s(0)e^{-\alpha k''_p z}[C_1 I_\alpha(\beta e^{-k''_p z}) + C_2 K_\alpha(\beta e^{-k''_p z})], \quad (2)$$

in which we have introduced two constants $\alpha = \frac{1}{2}$ and $\beta = \frac{\omega_s^2}{k''_p k_s c_0^2}\left|\frac{\chi_{eff}^{(2)} A_p}{k_s}\right|$. Here, $I_\alpha(\cdot)$ and $K_\alpha(\cdot)$ are the $\alpha$th-order modified Bessel functions of the first and second kinds respectively, and $C_1$ and $C_2$ are two constants to be determined by the initial conditions at $z = 0$.

Detailed processes in solving the partial differential equation above is given in Supplementary Note 1. At the transmission side, the signal gain (i.e., the ratio between the square of the electric field amplitudes at the waveguide's output ($z = L$) and input ports ($z = 0$)) are obtained as:

$$G = \left| \frac{\left(\frac{1}{2} + i\frac{\omega_s^2 \chi_{eff}^{(2)} |A_p|}{k''_p k_s c_0^2} e^{i\varphi_p - 2i\varphi_s}\right)\left[K_{\frac{1}{2}}(\beta)I_{\frac{1}{2}}\left(\beta e^{-k''_p L}\right) - I_{\frac{1}{2}}(\beta)K_{\frac{1}{2}}\left(\beta e^{-k''_p L}\right)\right]}{+ \beta\left[K_{\frac{1}{2}}'(\beta)I_{\frac{1}{2}}\left(\beta e^{-k''_p L}\right) - I_{\frac{1}{2}}'(\beta)K_{\frac{1}{2}}\left(\beta e^{-k''_p L}\right)\right]}{\beta\left[I_{\frac{1}{2}}(\beta)K_{\frac{1}{2}}'(\beta) - I_{\frac{1}{2}}'(\beta)K_{\frac{1}{2}}(\beta)\right]} \right|^2 e^{-(k''_p + 2k''_s)L}$$

$$(3)$$

From Eq. (3), we note that the nonlinear gain depends on the incident pump-wave amplitude $|A_p|$, the phase difference between the incident pump and signal waves $\varphi_p - 2\varphi_s$, but is independent of the loss tangent. In other words, the signal wave can be modulated from distinct amplification to perfect absorption by tuning the pump-wave intensity and the phase difference between the input waves (see Fig. 1d). The increase of pump energy yields larger maximum gain and absorption rate of the signal wave. In addition, the existence of transmission loss leads to an upper limit of the signal gain with respect to the transmission length $L$.

**Design of nonlinear SSPP waveguide**

Figure 2 gives the schematic diagram of the nonlinear SSPP waveguide to dynamically control the signal wave from CPA to amplification. The geometric parameters of the nonlinear SSPP unit cell are shown in Fig. 2b, where $d_1 = 2.5$ mm, $d_2 = 3$ mm, $d_3 = 1$ mm, $h_1 = 3$ mm, $h_2 = 12$

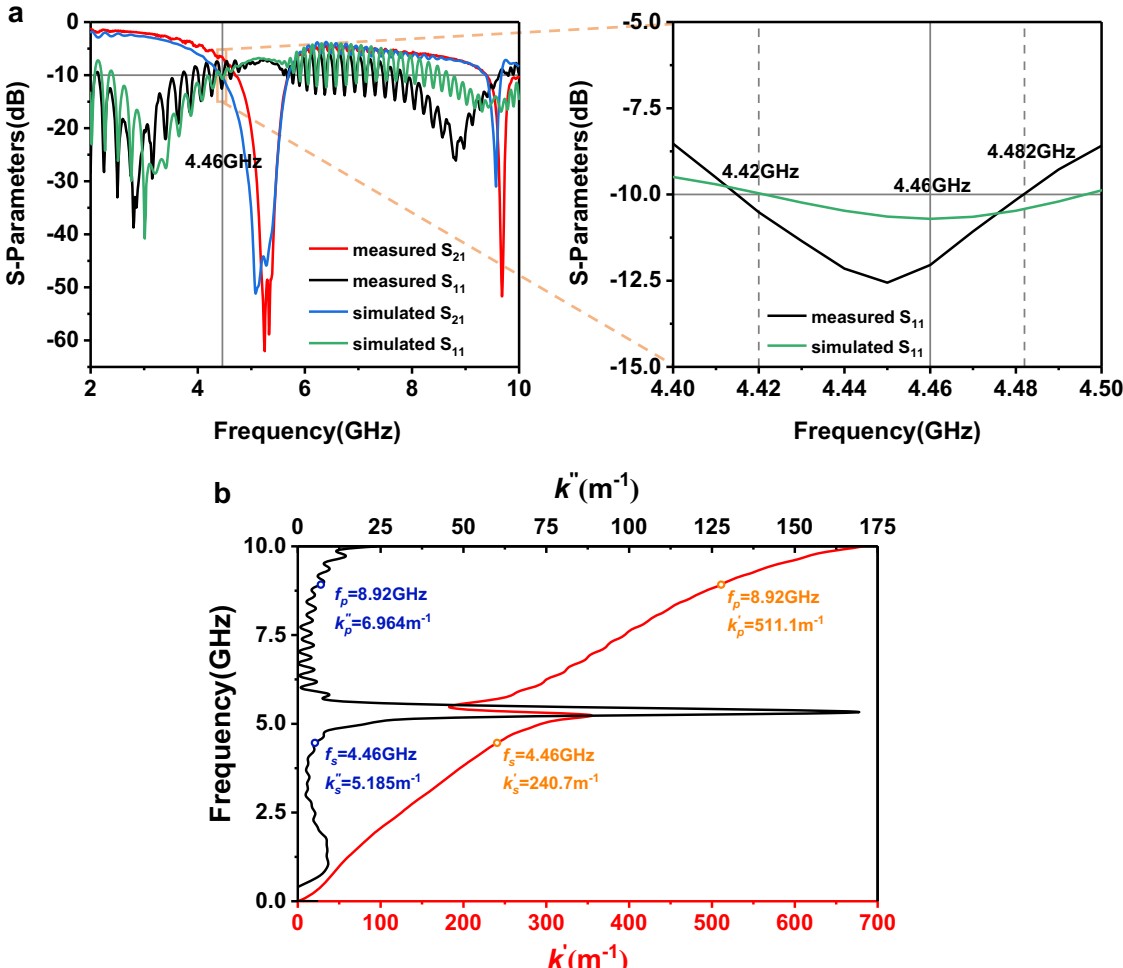

**Fig. 3 | The S-parameters and dispersion relation of the spoof surface plasmon polariton (SSPP) waveguide. a** The simulated and measured S-parameter curves of the SSPP waveguide with the unit number $N = 60$. **b** The calculated dispersion and loss relations of the SSPP waveguide.

mm, $h_3 = 3$ mm, $p = 4$ mm, $w = 2.7$ mm, $a = 1$ mm, $s = 0.3$ mm, $x_1 = 3$ mm, and $x_2 = 6$ mm. The substrate material is Rogers RO4350B with the thickness $t = 0.508$ mm, relative permittivity $\varepsilon_r = 3.66$, and loss tangent $\tan\delta = 0.0037$. Two through-holes are punched on each unit cell to conduct currents to the ground plane, and a varactor diode (MAVR-011020-1411) is loaded in the gap of each unit cell. A low-power signal wave and a high-power pump wave passing through a phase shifter are jointly injected into the nonlinear SSPP waveguide, where the relative phase between the signal and pump waves is dynamically tuned to switch the output signal wave from attenuated to amplified states, as depicted in Fig. 2c.

**Simulations and measurements**

We simulate the reflection and transmission coefficients of the nonlinear waveguides using the commercial software, CST Microwave Studio. Figure 3a presents the simulated and measured S-parameter results of the SSPP waveguide with the unit number $N = 60$, where the reflection coefficient is smaller than −10 dB around the phase matching point 4.46 GHz (meaning that the reflected energy is less than 10%). The dispersion and loss relations are obtained through the generalized Bianco-Parodi method[49,50], as shown in Fig. 3b. With zero bias voltage, the varactor diode has the resistance $R = 13.5\ \Omega$ and capacitance $C = 0.23$ pF, which are used in the simulations. According to the calculated dispersion curves, when the measured signal wave frequency $f_s \approx 4.46$ GHz and the pump wave frequency $f_p = 2f_s = 8.92$ GHz (corresponding to the calculated real

parts $k_s' = 240.7\text{m}^{-1}$ and $k_p' = 511.1\text{m}^{-1}$ of the wave numbers, respectively), the phase matching condition of $k_p' = 2k_s'$ is roughly fulfilled (with 5.8% error of $k_p'$ induced by the generalized Bianco-Parodi method).

Figure 4a shows the experimental setup, in which the signal and pump waves are provided by two signal generators, respectively. A phase shifter is applied to adjust the phase of the signal wave to change the phase relation between the input waves ($\varphi_p - 2\varphi_s$). The modulated signal wave and pump wave are coupled into the nonlinear SSPP waveguide through a coupler, and the output waves are received and analyzed by a spectrum analyzer.

As discussed in theoretical analysis, under the condition of $f_p = 2f_s$, the differential frequency wave generated from three-wave mixing process has the same frequency with the signal wave, and their positive interference will result in significant gain of the signal wave. When the signal and differential frequency waves are out of phases, the two waves will interfere and cancel each other, leading to total absorption of the signal wave. The phase relation $\varphi_p - 2\varphi_s$ corresponding to the PA and CPA cases can be derived from the three-wave mixing equation:

$$\frac{\mathrm{d}A_i(z)}{\mathrm{d}z} = \mathrm{i}\frac{\omega_i^2}{k_i c_0^2}\chi_{\text{eff}}^{(2)}A_p(z)A_s^*(z)e^{-\mathrm{i}\Delta k'z}e^{-k_p''z} \tag{4}$$

where $A_i$, $\omega_i$ and $k_i$ are respectively the electric field amplitude, frequency and wave number of the idler wave (differential frequency

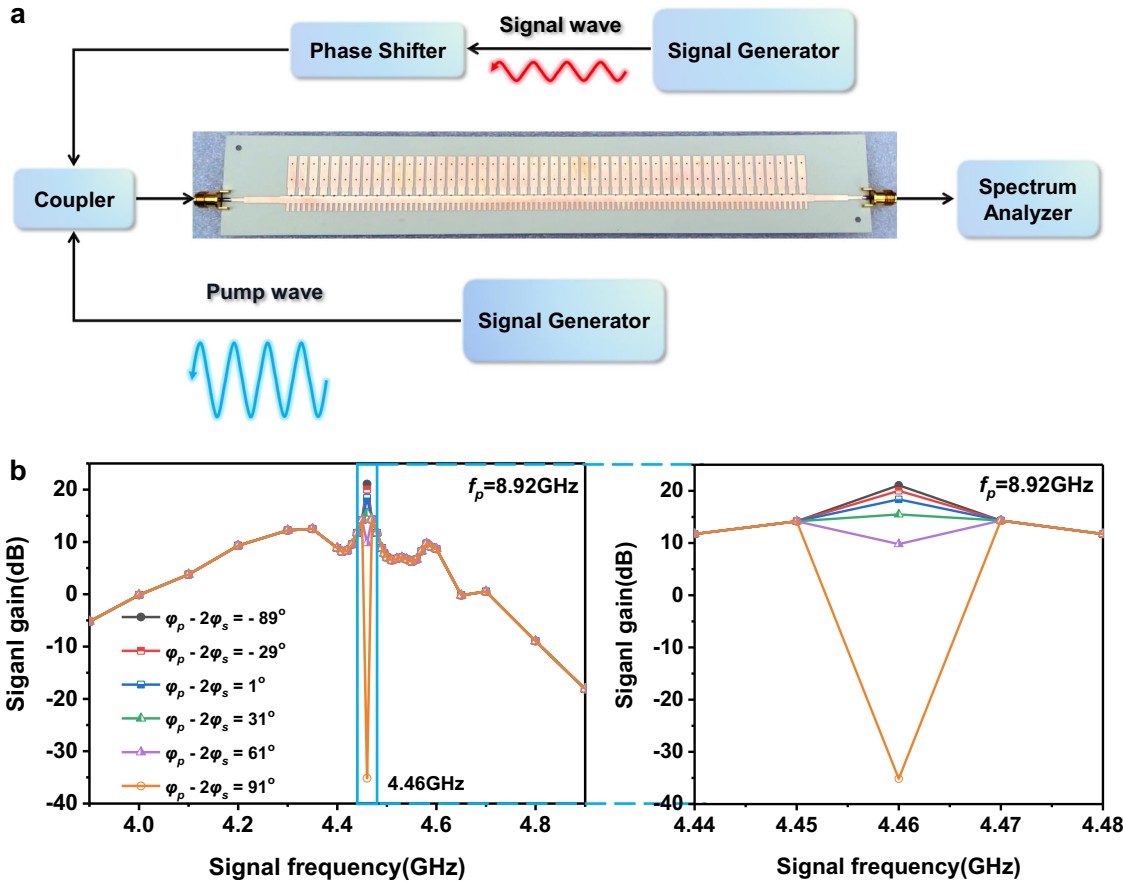

**Fig. 4 | The measured signal gains of the spoof surface plasmon polariton (SSPP) waveguide. a** The diagram of experimental setup. **b** The measured signal gains versus the signal frequencies with different input phase relations $\varphi_p$-$2\varphi_s$ at the fixed pump frequency $f_p = 8.92$ GHz. The unit number of the SSPP waveguide is $N = 60$.

wave). In our degenerate cases, $\omega_i = \omega_s$, $k_i = k_s$ and $\Delta k' = 2k'_s - k'_p = 0$. $k_s$ is a complex number so the factor $\arg(k_s)$ also affects the phase. The phases on both sides of the equation are also equal, i.e.,

$$\varphi_i = 90^\circ - \arg(k_s) + \varphi_p - \varphi_s \qquad (5)$$

So $\varphi_i - \varphi_s = (\varphi_p - 2\varphi_s) + 90^\circ - \arg(k_s)$. When $\varphi_i - \varphi_s = 0^\circ$ and $180^\circ$, which correspond to the maximum and minimum gain respectively, $\varphi_p - 2\varphi_s = -90^\circ + \arg(k_s)$ and $90^\circ + \arg(k_s)$. In our case, $\arg(k_s) \approx 1.234^\circ$, but due to the limitation of the phase adjustment accuracy in the experiment, we can only set $\arg(k_s) = 1^\circ$.

Figure 4b shows the measured signal gains versus $f_s$ under different input phase relations when the nonlinear SSPP waveguide has 60 units and $f_p$ is fixed to 8.92 GHz. It can be observed that only at $f_s = \frac{1}{2}f_p = 4.46$ GHz, the gain of the signal wave is affected by the input phase relation, and a sharp peak and a deep dip are observed at $\varphi_p - 2\varphi_s = -89^\circ$ and $91^\circ$, respectively. This single frequency behavior results from the abrupt switching between degenerate and non-degenerate three-wave mixing process. However, if we fix the relationship of $f_p = 2f_s$, instead of fixing $f_p$, there will be a small operation linewidth (see Supplementary Fig. 1 in the Supplementary Information).

Equation (3) suggests that the signal gain is related to the attenuation constants ($k''_s$ and $k''_p$) of the input waves and the transmission distance (or the length $L = Np$ of the SSPP waveguide). In the lossless case, the signal gain of parametric amplification will be enhanced continuously by increasing the transmission distance. However, the varactor diodes embedded in the SSPP structures have non-negligible loss, which accumulates with the increase of the waveguide length. Hence, the signal gain does not change

monotonously with the propagation length and there is an optimal waveguide length that can maximize the signal gain. Selecting appropriate varactor diodes and substrates can help obtain good robustness of the optimal waveguide length (the detailed discussions are given in Supplementary Note 3). We have tested the signal gains for the SSPP waveguides of 50, 60, 70, 80, and 110 units to find the optimal value of waveguide length. The signal gain at $f_s = \frac{1}{2}f_p = 4.46$ GHz is theoretically calculated as a function of the propagation distance ($L/\lambda$), as shown in Fig. 5a. In the experiment, we measured the signal gain of SSPP waveguides with 50, 60, 70, 80, and 110 units, corresponding to 7.7, 9.2, 10.7, 12.3, and 16.9 SSPP signal wavelengths, respectively. The input pump power is 21.52 dBm, calculated as the power provided by the signal generator minus the losses of the coupler and coaxial lines. The input signal power will not affect the signal gain, but it cannot be very large to meet the precondition in the non-depletion pump regime in the theoretical derivation[22]. The highest signal power that can still achieve the maximum modulation depth of the device is −25 dBm.

The measured signal gains at different transmission distances are also plotted in Fig. 5a, showing excellent agreement with the theoretical calculations. When $\varphi_p - 2\varphi_s = -89^\circ$, the signal gain will take the maximum value, and the peak gain of 21.98 dB can be measured when the waveguide has 60 units (about 9.2 wavelengths). When $\varphi_p - 2\varphi_s$ increases, the signal gain will gradually decrease until $\varphi_p - 2\varphi_s = 91^\circ$, where it will be attenuated to the minimum, about −33 dB (more than 90% absorption after deducting the reflection). In Fig. 5b, the signal gain versus the phase difference $\varphi_p - 2\varphi_s$ for different propagation lengths is further discussed. We can observe that as the phase difference increases, the signal gain first declines slowly but drops rapidly after the phase difference is larger than $31^\circ$. When the phase difference

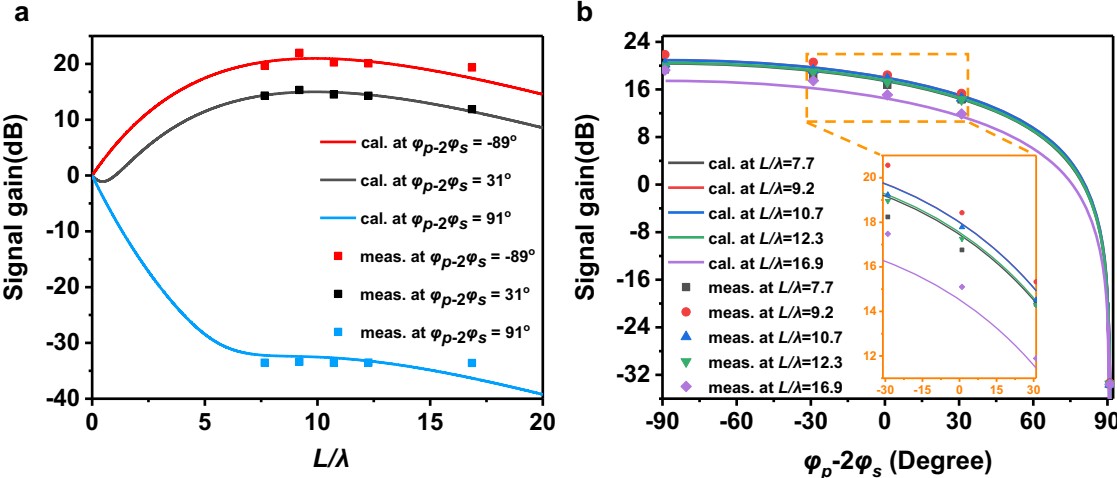

**Fig. 5 | The calculated and measured signal gains of the nonlinear spoof surface plasmon polariton waveguide. a** The signal gains versus the propagation length under distinct phase differences $\varphi_p - 2\varphi_s$. **b** The signal gains versus the phase difference between the signal and pump waves under different transmission distances $L/\lambda$.

reaches about 91°, the signal wave is attenuated to the minimum, almost completely absorbed. As the phase difference further increases, the signal gain will rise back to the maximum value in a mirror symmetric trend. The measured modulation range of the nonlinear SSPP waveguide (from −33 dB to 22 dB) can be further increased if higher input power of the pump wave can be provided (detailed discussions are given in Supplementary Note 4).

## Discussion

Coherent perfect absorption is a phenomenon of complete absorption achieved by controlling the interference of multiple incident waves, which is of great interest for many applications such as biological sensing, photo detection, radar stealth, solar-thermal energy sharing, and imaging. Meanwhile, parametric amplification is a nonlinear process, where the signal can be amplified by a pump wave via the generation of an idler wave. In this work, we theoretically and experimentally demonstrate both CPA and PA at the same frequency in a single nonlinear SSPP waveguide. More importantly, the nonlinear SSPP waveguide can also provide dynamic switching of the signal wave from complete extinction to significant amplification, by simply tuning the relative phase between the signal and pump waves. The proposed method may find potential applications in on-chip sensing, imaging, and wireless communication systems.

## Methods
### Numerical simulations

The simulated curves of scattering parameters are obtained using the time domain solver of Commercial Software, CST Microwave Studio. The nonlinear SSPP waveguide is placed in the open boundaries and motivated by the waveguide ports. We use the lumped element with area equaling to the dimension of varactor diode to imitate the varactor diode, and the type, resistance, inductance and capacitance of the lumped element are RLC serial, 13.5Ω, 0H, and 0.23 pF, respectively. Since the dimension of the unit structure is large enough relative to the varactor diode, we do not need to model the varactor.

### Waveguide fabrication

Our nonlinear waveguide is composed of a 0.018 mm-thick copper SSPP layer, a 0.508 mm-thick Rogers RO4350B substrate layer, and a 0.018 mm-thick copper ground. The SSPP strip and the copper ground are connected by two through-holes with radius $r = 0.2$ mm. The structural fabrication error is within ±0.03 mm.

### Measurement

The low-power signal wave and high-power pump wave are provided by two signal generators with the models of Agilent E8257D-520 and E8257D-540, respectively. We adopt the phase shifter (Qualwave QMPS90) to adjust the phase of the signal wave, and the modulated signal wave and pump wave are coupled by a coupler (Midwest Microwave, CPL-5231-16-001-79) to input together to the nonlinear SSPP waveguide. Finally, the output waves are monitored by a spectrum analyzer (Agilent N9040B). Moreover, the spectrum analyzer and two generators are connected with the BNC lines for synchronization.

## Data availability

The authors declare that all relevant data are available in the paper and its Supplementary Information files, or from the corresponding author on request.

## Code availability

The custom computer codes used in this study are available from the corresponding authors on request.

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

## Acknowledgements

This work was supported by the National Natural Science Foundation of China (62271139, U21A20459, 61871127 J.Z., 62288101 T.J.C.), the National Key Research and Development Program of China (2022YFA1404903 J.Z.), the Distinguished Professor Fund of Jiangsu Province (Grant No. 1004-YQR24010 Y.L.), the 111 Project (111-2-05 T.J.C.), and Postgraduate Research& Practice Innovation Program of Jiangsu Province (3304002304D W.Y.C.).

## Author contributions

J.Z., Y.L. and T.J.C. suggested the designs and planned and supervised the work. Y.L. performed theoretical analysis. W.Y.C. and X.G. carried out the numerical simulations and experimental measurements. W.Y.C., J.Z., Y.L. and T.J.C. wrote the manuscript.

## Competing interests

The authors declare no competing interests.
