## [Peer Review File · Nature Communications]

REVIEWER COMMENTS

Reviewer #1 (Remarks to the Author):

This work demonstrates a new scheme for dynamic switching from parametric amplification (PA) to coherent perfect absorption (CPA) in a nonlinear spoof plasmonic waveguide by using three-wave mixing process. To my knowledge, such scheme is novel and has not been reported. The signal gain at the transmission side can be tuned from -33dB to 22dB by readily tailoring the phase difference between the pump and the signal. The signal gain is also expected to be tuned continuously when an analog phase shifter is employed. Experimental results show good agreements with the calculated results. Both the idea and the experimental realization are interesting, and the manuscript is well organized. I feel that it merits publication in Nature Communications after the authors address the following comments:

1) The concept of nonlinear CPA in a two-port system is not intuitively given. In this work, the signal gain is used as the exclusive parameter to define CPA. However, different from the one-port system (shown in Fig. 1(a)) where $R+A=1$, in the two-port system, zero transmission doesn't equal to unity absorption. As in Fig. 3(a), only S_{21} is given, yet S_{11} in such system is more likely to be non-zero. Thus, the authors are suggested to calculate and measure (if possible) the reflection for all cases (from Fig. 3 to Fig. 5).

2) It would be better to give the nonlinear CPA condition step by step, if possible. From Eq. (3), it is not intuitive that how to obtain the CPA condition of $\phi_p - 2\phi_s = 90^\circ$. Is it only satisfied for some specific $\chi_{\text{eff}}^2 * |A_p|$, or for arbitrary $\chi_{\text{eff}}^2 * |A_p|$? For example, if $\chi_{\text{eff}}^2 * |A_p|$ increases or decreases 10 times, will the gain at the CPA frequency significantly change?

3) Traditional linear CPA can only be achieved at some fixed resonance conditions, i.e. once the frequency is fixed, the CPA phenomenon can only be observed for some discrete cavity sizes. On the contrary, Fig. 5a shows that the nonlinear CPA is quite different as it is quite robust to the waveguide geometry once the waveguide length is larger than a critical value. I believe that this point is very interesting and hence deserves more discussions. For instance, what parameter(s) will affect the critical length of the waveguide, above which the gain/absorption coefficient approaches a constant.

4) The theoretical model developed to describe the PA and CPA is based on undepletion pump approximation, which requires the amplitude of the signal wave to be sufficiently small compared with that of the pump wave. I am wondering what the maximum input signal power the nonlinear waveguide is able to amplify/absorb.

5) The maximum gain appears when $\phi_p - 2\phi_s = -90^\circ$, while the minimum gain emerges when $\phi_p - 2\phi_s = 90^\circ$. By substituting the phase matching condition of the three-wave mixing process ($\phi_p = \phi_s + \phi_i$) into them, one can get that $\phi_i = \phi_s - 90^\circ$ and $\phi_i = \phi_s + 90^\circ$, respectively. Why does the constructive and destructive interference occur when the phase difference between the signal wave and the generated differential wave reaches -90° and 90° , instead of 0° and 180° ?

6) In the manuscript, the authors state that the increase of pump energy yields larger maximum gain and absorption rate of the signal wave. Please give more details (a quantitative study) on this point.

Minor points:

1) Fig. 1(d), the y-axis denotes ϕ_s , however, CPA and PA depend on $\phi_p - 2\phi_s$.

2) Fig. 4(a), the arrows of the signal wave and the pump wave seem to go backwards. Fig. 4(b), the dip and peaks are not clearly seen.

3) Fig. 5(a), it seems that perfect absorption occurs when the propagation length is larger than 7 wavelengths. However, on page 4, line 80, it claims that "it is insensitive to the material's loss tangent and dimensions of the device (e.g. the length of SSPP waveguide). The statement should be re-considered.

4) The legend of the horizontal axis of Fig. 5a is "Propagation length (wavelength number)". The "wavelength number" here can easily get people confused with "wave number". It is better to rephrase the legend here, for example, to "Normalized waveguide length" or simply " $L/\text{wavelength}$ " to avoid confusion.

5) I find that a pioneering work in the field of CPA is missing in the reference: Y. D. Chong, Li Ge, Hui Cao, and A. D. Stone, Phys. Rev. Lett. 105, 053901

Reviewer #2 (Remarks to the Author):

In this manuscript Cui and co-workers present a very interesting design able to achieve both coherent perfect absorption and parametric amplification at the same frequency in the same device. The design is based on a corrugated thin waveguide that supports the propagation of spoof surface plasmons in which a varactor diode is introduced to act as a nonlinear generator. The manuscript is very well written, the results are interesting and relevant, and the main conclusions regarding the feasibility of using this type of dynamic switchers is supported by both the experimental findings and the background theory. Also, the agreement between the numerical simulations and the experimental results is indeed very good. I do not have any objection on the scientific quality and depth of the results. Therefore, in principle, I am inclined to support the publication of this manuscript in Nature Communications.

My only major suggestion is related to incorporating a more detailed discussion on the operation linewidth of the device, as shown in Figure 4. First, it would be instructive to analyse the physical mechanism behind the gain to understand why is maximum near the frequency of operation (4.43 GHz). Second, it would be great to know why there is only one point leading to complete absorption. In every resonant phenomenon, there is always a linewidth that is associated with a limiting physical magnitude. What is the linewidth and the associated limiting factor in this case? I think that this discussion could add more physical insight into this proposal. In addition, as in the title the word "dynamic" has been incorporated, it would be interesting to discuss the timescale of the dynamical switching present in this type of devices.

Reviewer #3 (Remarks to the Author):

The authors proposed and realized a dynamic switching mechanism from coherent perfect absorption to parameter amplification by using a nonlinear spoof plasmonic waveguide. The idea is very interesting. The results are validated by both simulations and microwave experiments. The device is a compact structure that could find useful applications in the future. Therefore, I highly recommend publishing this paper on Nat. Commun., and I only have some minor suggestions for the authors to improve the manuscript.

1. The mechanism of nonlinear process is elaborated by Eqs. 1-3. But can the authors describe briefly in a physical way why they choose this structure for nonlinear process and is it optimized?

2. It seems from Fig. 2 that the amplification or attenuation is applied to the signal wave, instead of the pump wave. I wonder if it is possible to apply to the pump wave instead. In other words, what happens if the signal wave has a much larger amplitude than the pump wave? Or in general, is it possible to control a wave of larger amplitude via a wave of smaller amplitude, like a triode.

3. What happens if the signal and pump waves have similar amplitudes. Usually what is the criteria on the ratio between them?

4. Will the structure of effective nonlinear media induce reflection? I understand that here the input port may reflect the reflected wave back into the structure again, but in this way, the length between the port and the structure might be important and worth mentioning. Since the absorption is discussed, it is better to make a clearer clarification.

5. I notice that the authors may have missed some related important papers of CPA in the reference list, e.g. *Physical Review A* 94, 063841 (2016), *EPL* 114, 28003 (2016), *Laser & Photonics Reviews* 12, 1800001 (2018), *Applied Physics Letters* 120, 171703 (2022), *Science* 377, 995 (2022). I suggest the authors add them and discuss appropriately.

Replies to the Referees' Comments

We appreciate the referees' constructive comments and suggestions very much, which help improve the quality of the manuscript significantly. Based on these comments and suggestions, we have revised the manuscript carefully, and the correction parts are highlighted in the yellow background. Below are our point-by-point replies (in blue fonts) to the referees' comments (in black fonts).

To Referee #1

Comment:

This work demonstrates a new scheme for dynamic switching from parametric amplification (PA) to coherent perfect absorption (CPA) in a nonlinear spoof plasmonic waveguide by using three-wave mixing process. To my knowledge, such scheme is novel and has not been reported. The signal gain at the transmission side can be tuned from -33dB to 22dB by readily tailoring the phase difference between the pump and the signal. The signal gain is also expected to be tuned continuously when an analog phase shifter is employed. Experimental results show good agreements with the calculated results. Both the idea and the experimental realization are interesting, and the manuscript is well organized. I feel that it merits publication in Nature Communications after the authors address the following comments:

Our Reply:

Thank you very much for your positive comments.

Comment:

1) The concept of nonlinear CPA in a two-port system is not intuitively given. In this work, the signal gain is used as the exclusive parameter to define CPA. However, different from the one-port system (shown in Fig. 1(a)) where $R+A=1$, in the two-port

system, zero transmission doesn't equal to unity absorption. As in Fig. 3(a), only S_{21} is given, yet S_{11} in such system is more likely to be non-zero. Thus, the authors are suggested to calculate and measure (if possible) the reflection for all cases (from Fig. 3 to Fig. 5).

Our Reply:

Thank you very much for raising this point. In fact, this paper focuses on two-port cases with good impedance matching and negligible reflection. To simplify the theoretical analysis, we suppose that the two-port system is non-reflective, where $T+A=1$. In our experiment, we designed an SSPP waveguide which has very small reflection. As shown in Figure R1, the measured reflection coefficient of the SSPP waveguide is smaller than -10 dB (meaning that the reflection is less than 10%) around the frequency of phase matching point 4.46 GHz, indicating that our case can be approximated as non-reflective, and the maximum absorption rate (when the measured gain is -33dB) is higher than 90%. Thus, the calculation results from our simplified theoretical model agree well with our experiments. To make this point clearer, we add the simulated and measured S_{11} parameters in Figure 3a (see Figure R1 below), and point out in the revised manuscript that our theoretical model applies only to ideal non-reflective cases. We would also like to highlight that, in contrast to a two-port linear system which is unable to realize CPA without reflection, our nonlinear CPA approach does not rely on the interference between the incident and reflected waves and can be achieved in a nonreflective system.

Figure R1 | The simulated and measured S-parameters of the SSPP waveguide.

Comment:

2) It would be better to give the nonlinear CPA condition step by step, if possible. From Eq. (3), it is not intuitive that how to obtain the CPA condition of $\varphi_p - 2\varphi_s = 90^\circ$. Is it only satisfied for some specific $\chi_{\text{eff}}^2 * |A_p|$, or for arbitrary $\chi_{\text{eff}}^2 * |A_p|$? For example, if $\chi_{\text{eff}}^2 * |A_p|$ increases or decreases 10 times, will the gain at the CPA frequency significantly change?

Our Reply:

Thank you very much for your question. Our nonlinear CPA method is enlightened by the difference frequency generation (DFG) of three-wave mixing. The principle is that the idler wave generated by the DFG process interferes with the signal wave and completely cancels it at a specific phase, which requires $f_p = 2f_s$. The detailed derivation of the nonlinear CPA is given in the supporting information. The phase relation of the three waves can be deduced from the nonlinear coupled equation:

$$\frac{dA_i(z)}{dz} = i \frac{\omega_i^2}{k_i c_0^2} \chi_{\text{eff}}^{(2)} A_p(z) A_s^*(z) e^{-i\Delta k'z} e^{-k_p''z},$$

in which $\omega_i = \omega_s$, $k_i = k_s$. Since k_s in the denominator is a complex number, the factor $\arg(k_s)$ also affects the phase. The phases on both sides of the equation are also equal, i.e.

$$\varphi_i = 90^\circ - \arg(k_s) + \varphi_p - \varphi_s - \Delta k'z.$$

Thus,

$$\varphi_i - \varphi_s = (\varphi_p - 2\varphi_s) + 90^\circ - \arg(k_s) - \Delta k'z.$$

When $\Delta k' \neq 0$, no matter how we set the initial phase difference $\varphi_p - 2\varphi_s$, the phase relation of the signal and idler waves $\varphi_i - \varphi_s$ will change during propagation. The PA (CPA) requires the phase difference between the idler wave and the signal wave always equal to 0 (π) so the nonlinear PA (CPA) process also requires the phase-matching condition $\Delta k' = 2k_s' - k_p' = 0$. In conclusion, the nonlinear conditions for PA

and CPA are $f_p = 2f_s$ and $k'_p = 2k'_s$, and the PA and CPA actually occur at $\varphi_p - 2\varphi_s = -90^\circ + \arg(k_s)$ and $90^\circ + \arg(k_s)$, respectively. For our structure, $\arg(k_s) \approx 1.234^\circ$.

To investigate how $\chi_{\text{eff}}^{(2)} |A_p|$ affects the signal gain, we plot the signal gain in terms of $\varphi_p - 2\varphi_s$ for different $\chi_{\text{eff}}^{(2)} |A_p|$, as shown in Figure R2. We observe that the maximum signal gain, which always occurs at $\varphi_p - 2\varphi_s = -90^\circ + \arg(k_s)$, increases with $\chi_{\text{eff}}^{(2)} |A_p|$. And similarly, the maximum attenuation which is always achieved at $\varphi_p - 2\varphi_s = 90^\circ + \arg(k_s)$ also increases with the value of $\chi_{\text{eff}}^{(2)} |A_p|$.

Figure R2 | The signal gains versus the phase difference $\varphi_p - 2\varphi_s$ with transmission distance $L = 2.5\lambda$.

Comment:

3) Traditional linear CPA can only be achieved at some fixed resonance conditions, i.e. once the frequency is fixed, the CPA phenomenon can only be observed for some discrete cavity sizes. On the contrary, Figure 5a shows that the nonlinear CPA is quite

different as it is quite robust to the waveguide geometry once the waveguide length is larger than a critical value. I believe that this point is very interesting and hence deserves more discussions. For instance, what parameter(s) will affect the critical length of the waveguide, above which the gain/absorption coefficient approaches a constant.

Our Reply:

Thank you very much for your good suggestion. In an ideal lossless case, the gain of the signal wave will increase monotonically with the propagation length under the phase matching condition in the PA cases. However, in the presence of the loss, there is a critical length at which the gain reaches the peak value. Note that the gain remains relatively stable (within 2dB variation) around the peak in a certain range of propagation length, as highlighted in Figures R3(a), (c) and (e). As the propagation length further increases, the loss will dominate and the total gain will gradually decrease. So the loss tangent determined by the substrate material and nonlinear elements will affect k_s'' and k_p'' , and thus the critical length as well as the robustness.

In addition, the $\chi_{\text{eff}}^{(2)}$ determined by the nonlinear elements also has a direct effect on the critical length and robustness. Here, we plot the signal gain as the function of the propagation length L , and investigate how k_s'' , k_p'' , and $\chi_{\text{eff}}^{(2)}$ would affect the critical length, as shown in Figure R3. In the PA case, we can observe that the maximum signal gain as well as the critical length decrease with larger k_s'' or k_p'' , but increase with $\chi_{\text{eff}}^{(2)}$.

Meanwhile, the attenuation increases monotonically with the propagation length in the CPA cases, as shown in Figure R3. Here, we mark the critical length at which the attenuation reaches 30 dB (corresponding to 99.9% attenuation). Note that the critical length for attenuation decreases with the increase of k_s'' or $\chi_{\text{eff}}^{(2)}$, but increases

with k_p'' . We have added these discussions on Pages 6-7 of the revised supplementary information.

Figure R3 | The signal gains versus the propagation length (L/λ) with different (a-b) k_s'' , (c-d) k_p'' and (e-f) $\chi_{\text{eff}}^{(2)}$ in (a,c,e) PA ($\varphi_p - 2\varphi_s = -90^\circ + \arg(k_s)$) and (b,d,f) CPA ($\varphi_p - 2\varphi_s = 90^\circ + \arg(k_s)$) cases.

Comment:

4) The theoretical model developed to describe the PA and CPA is based on undepletion pump approximation, which requires the amplitude of the signal wave to

be sufficiently small compared with that of the pump wave. I am wondering what the maximum input signal power the nonlinear waveguide is able to amplify/absorb.

Our Reply:

Thank you very much for your good question. From Eq. (3), we note that the signal power has no influence on the signal gain under the undepletion pump approximation. However, as the signal power increases, the undepletion approximation is no longer valid, and the measured signal gain deviates from the calculated results, as shown in Figure R4. Here, we plot the calculated and measured signal gains as functions of the signal power at $\varphi_p - 2\varphi_s = -90^\circ + \arg(k_s) \approx 61^\circ, 31^\circ$ and -89° when the pump power is 21.59dBm. We observe that the signal gain will finally fall as the signal power reaches a certain value. In order to achieve the maximum signal gain, the signal power needs to be below -25dBm. In other words, for the undepleted pumped approximation to work, the signal wave should remain 4 order of magnitude smaller than the pumped one. However, we would like to highlight that beyond the undepleted pump approximation, our approach still works with a downgraded performance. These discussions have been added on Page 7-8 of the revised supplementary information.

Figure R4 | The signal gains versus the signal power.

Comment:

5) The maximum gain appears when $\varphi_p - 2\varphi_s = -90^\circ$, while the minimum gain emerges when $\varphi_p - 2\varphi_s = 90^\circ$. By substituting the phase matching condition of the three-wave mixing process ($\varphi_p = \varphi_s + \varphi_i$) into them, one can get that $\varphi_i = \varphi_s - 90^\circ$ and $\varphi_i = \varphi_s + 90^\circ$, respectively. Why does the constructive and destructive interference occur when the phase difference between the signal wave and the generated differential wave reaches -90° and 90° , instead of 0° and 180° ?

Our Reply:

Thank you for your question. For the three-wave mixing process, the nonlinear coupled equation is given by:

$$\frac{dA_i(z)}{dz} = i \frac{\omega_i^2}{k_i c_0^2} \chi_{\text{eff}}^{(2)} A_p(z) A_s^*(z) e^{-i\Delta k' z} e^{-k_p^* z}$$

In our degenerate cases, $\omega_i = \omega_s$, $k_i = k_s$ and $\Delta k' = 0$. k_s is a complex number so the factor $\arg(k_s)$ also affects the phase. The phases on both sides of the equation are also equal, i.e.

$$\varphi_i = 90^\circ - \arg(k_s) + \varphi_p - \varphi_s$$

Thus,

$$\varphi_i - \varphi_s = (\varphi_p - 2\varphi_s) + 90^\circ - \arg(k_s)$$

When $\varphi_p - 2\varphi_s = -90^\circ (90^\circ) + \arg(k_s)$, $\varphi_i - \varphi_s = 0^\circ (180^\circ)$, and the idler wave generated by the DFG process interferes with the signal wave and amplifies/completely cancels it to realize PA/CPA. We have added these discussions on Page 9 of the revised manuscript.

Comment:

6) In the manuscript, the authors state that the increase of pump energy yields larger maximum gain and absorption rate of the signal wave. Please give more details (a quantitative study) on this point.

Our Reply:

Thank you for your suggestion. Here, we calculate and measure the signal gain versus $\varphi_p - 2\varphi_s$ with different pump power, as shown in Figure R5. We can observe that the signal gain is very sensitive to the pump power. The lower the pump power is, the smaller the maximum gain and absorption will be. Theoretically, a 2dB decrease in pump power can reduce both the maximum gain and attenuation by about 6dB. These discussions can be found on Page 8 of the revised supplementary information.

Figure R5 | The signal gains versus $\varphi_p - 2\varphi_s$ with different pump power.

Minor points:

Comment:

1) Fig. 1(d), the y-axis denotes φ_s , however, CPA and PA depend on $\varphi_p - 2\varphi_s$.

Our Reply:

Thank you for pointing it out. As discussed above, CPA and PA depend on $\Delta\varphi = \varphi_p - 2\varphi_s - \arg(k_s)$, and the premise of Figure 1(d) is that the input phase of the pump wave φ_p is a fixed value. Following your suggestion, we have changed the y-axis in Figure 1(d) to $\varphi_p - 2\varphi_s$ in the revised manuscript, as shown in Figure R6.

Figure R6 | Principles of linear and nonlinear CPAs. (a) Schematic of a linear coherent perfect absorber consisting of a gain-loss medium layer on top of a metallic ground plane. The destructive interference between the waves reflected at the first and second interfaces results in the enhancement of absorption. (b) Gain/absorption of the linear system in terms of the thickness L and loss tangent of the gain-loss medium layer. CPA can be realized only at some specific slab thicknesses and with a fixed loss tangent of the absorbing material (c) Schematic of the nonlinear CPA. By controlling the phase difference between the input signal and pump waves, the interference between the transmitted signal and generated differential waves from a nonlinear medium can be tuned from constructive to destructive to achieve PA and CPA. (d) Gain/absorption of the signal wave in terms of the phase difference $\varphi_p - 2\varphi_s$ and the loss tangent of the nonlinear medium.

Comment:

2) Fig. 4(a), the arrows of the signal wave and the pump wave seem to go backwards. Fig. 4(b), the dip and peaks are not clearly seen.

Our Reply:

Thank you for your reminder. I have corrected the direction of the arrows of the signal wave and the pump wave, and added an inset showing zoomed-in dips and peaks. Ps: We performed the experiment again, and there is a slight deviation of measured f_s from the previous results (from 4.43 GHz to 4.46 GHz). The updated results are given in Figure 4 of the revised manuscript, as shown in Figure R7 below.

Figure R7 | (a) The diagram of experimental setup. (b) The measured signal gains versus the signal frequencies with different signal-wave phases at the fixed pump frequency $f_p = 8.92$ GHz.

The unit number of the SSPP waveguide is $N = 60$.

Comment:

3) Fig. 5(a), it seems that perfect absorption occurs when the propagation length is larger than 7 wavelengths. However, on page 4, line 80, it claims that “it is insensitive to the material’s loss tangent and dimensions of the device (e.g. the length of SSPP waveguide). The statement should be re-considered.

Our Reply:

Thank you for your correction. The statement here is not rigorous and we have revised it in the revised manuscript as: “it is insensitive to the material’s loss tangent and the CPA/PA effect can be achieved within a certain range of device dimensions”.

Comment:

4) The legend of the horizontal axis of Fig. 5a is “Propagation length (wavelength number)”. The “wavelength number” here can easily get people confused with “wave number”. It is better to rephrase the legend here, for example, to “Normalized waveguide length” or simply “ L/λ ” to avoid confusion.

Our Reply:

Thank you very much for your kind advice. I have unified the legend of the horizontal axis of the relevant figures in the revised manuscript to L/λ , as shown in Figure R8.

Figure R8 | The calculated and measured signal gains of the nonlinear SSPP waveguide. (a) The signal gains versus the propagation length under distinct phase differences $\phi_p - 2\phi_s$. (b) The signal gains versus the phase difference between the signal and pump waves under different transmission distances L/λ .

Comment:

5) I find that a pioneering work in the field of CPA is missing in the reference: Y. D. Chong, Li Ge, Hui Cao, and A. D. Stone, Phys. Rev. Lett. 105, 053901.

Our Reply:

Thank you for reminding us about this very relevant reference. We have added this work into the reference list in the revised manuscript.

To Referee #2**Comment:**

In this manuscript Cui and co-workers present a very interesting design able to achieve both coherent perfect absorption and parametric amplification at the same frequency in the same device. The design is based on a corrugated thin waveguide that supports the propagation of spoof surface plasmons in which a varactor diode is introduced to act as a nonlinear generator. The manuscript is very well written, the results are interesting and relevant, and the main conclusions regarding the feasibility of using this type of dynamic switchers is supported by both the experimental findings and the background theory. Also, the agreement between the numerical simulations and the experimental results is indeed very good. I do not have any objection on the scientific quality and depth of the results. Therefore, in principle, I am inclined to support the publication of this manuscript in Nature Communications.

Our Reply:

Thank you very much for your positive comments.

Comment:

1. My only major suggestion is related to incorporating a more detailed discussion on the operation linewidth of the device, as shown in Figure 4. First, it would be

instructive to analyse the physical mechanism behind the gain to understand why is maximum near the frequency of operation (4.43 GHz). Second, it would be great to know why there is only one point leading to complete absorption.

Our Reply:

Thank you very much for your good suggestion. As discussed in in the first paragraph of the theory section of the manuscript, our device is based on a three-wave-mixing process, where the generated wave (idler wave) has the same frequency of the signal wave. Thus, tuning the phase difference between the differential wave and the signal wave can result in destructive to instructive interference between them, leading to the control of signal wave from attenuation to amplification. The phase matching condition for this three-wave-mixing process ($f_p = 2f_s$ and $k'_p = 2k'_s$), which is critical to realizing highly efficient nonlinear conversion, can be achieved at a single frequency $f_s = 4.46$ GHz when the pump frequency is fixed at $f_p = 8.92$ GHz (We redid the experiment and there is a slight deviation of measured f_s). Second, because our pump frequency is fixed at $f_p = 8.92$ GHz, the maximum PA gain and complete absorption only occurs at a single frequency $f_s = 4.46$ GHz. At this frequency, we get degenerate three wave mixing process (where the idler and signal have the same frequency and thus interfere with each other), whereas, out of this frequency, we have nondegenerate three wave mixing process (where the idler and signal have different frequencies and thus do not interfere). The abrupt switching between degenerate and nondegenerate three wave mixing process leads to the single frequency behavior. On the other hand, if the pump frequency changes with the signal one as $f_p = 2f_s$, the signal gain (and/or the absorption ratio) will change smoothly with the frequency. Only in this case, we can define the operation linewidth.

Comment:

2. In every resonant phenomenon, there is always a linewidth that is associated with a limiting physical magnitude. What is the linewidth and the associated limiting factor in this case? I think that this discussion could add more physical insight into this proposal.

Our Reply:

Thank you very much for your good question. As discussed above, if we fix the pump frequency at the phase matching frequency point and vary the input signal frequency, we will only observe a single peak/dip of the signal gain. However, if we fix the relationship of $f_p = 2f_s$, instead of fixing f_p , there is indeed a linewidth (blue shaded region), as shown in Figure R9. The measured 3dB bandwidth is about 0.036GHz and the corresponding quality factor $Q = 2\pi \frac{4.46}{0.036} \approx 778$.

Figure R9 | The measured signal gains versus the signal frequencies at $f_p = 2f_s$ and $f_p = 8.92$ GHz. The unit number of the SSPP waveguide is $N = 60$.

The factors influencing the linewidth and the associated limiting factor are the phase mismatch $\Delta k' = 2k'_s - k'_p$ affected by the shape of the dispersion curve and the propagation length L . Here, we set $\Delta k' = \gamma \Delta \omega_s$, and vary the constant γ from $0.5\gamma_0$ to $1.5\gamma_0$ to qualitatively calculate the linewidth and Q versus the waveguide length under different $\Delta k'$. As shown in Figure R10, we verify that, as the phase mismatch or waveguide length increases, the operation linewidth decreases and the Q factor increases. Thus, to achieve a large operation linewidth, the dispersion curve must be well designed to keep the phase mismatch (out of the phase matching frequency) small and the optimal waveguide length short. We have added these discussions on pages 4-6 of the Supplementary Information.

Figure R10 | The calculated operation linewidth and Q factor versus the waveguide length L/λ in (a-b) amplification cases (c-d) and attenuation cases under different γ when $f_p = 2f_s$.

Comment:

In addition, as in the title the word ``dynamic'' has been incorporated, it would be interesting to discuss the timescale of the dynamical switching present in this type of devices.

Our Reply:

Thank you for your suggestion. Three-wave mixing is a transient optical effect, so the timescale of the dynamical switching depends primarily on the phase shifter. The switching speed of most of the digital phase shifter models is 100ns. We found that the switching speed of the model HWQDPH2060-6 is only 35ns, but it can only be used to tune the signal wave due to its operating frequency.

To Referee #3**Comment:**

The authors proposed and realized a dynamic switching mechanism from coherent perfect absorption to parameter amplification by using a nonlinear spoof plasmonic waveguide. The idea is very interesting. The results are validated by both simulations and microwave experiments. The device is a compact structure that could find useful applications in the future. Therefore, I highly recommend publishing this paper on Nat. Commun., and I only have some minor suggestions for the authors to improve the manuscript.

Our Reply:

Thank you very much for your positive comments.

Comment:

1. The mechanism of nonlinear process is elaborated by Eqs. 1-3. But can the authors describe briefly in a physical way why they choose this structure for nonlinear process and is it optimized?

Our Reply:

Thank you very much for your good question. Phase matching is critical to our approach, which requires precise control of the wave vectors of signal, pump, and idler waves. Spoof SPP platform give us more freedom to control the wave vectors and hence to design the phase matching. This is why we choose this platform.

Figure R11 | (a) The signal gains versus $\phi_p - 2\phi_s$ with different pump power. (b) The signal gains versus the input signal power

In addition to satisfying the phase matching conditions, the spoof SPP structure was optimized to achieve the following goals: 1) the electromagnetic field is strongly localized around the varactor diodes; 2) high S_{21} at f_p to obtain improved gain/attenuation (As shown Figure R11, the magnitude of the pump signal has critical influence on the gain/attenuation, whereas increasing the signal power will not improve the gain/attenuation); 3) as small λ_{SSPP} as possible to realize a compact device. Finally, we fabricated SSPP waveguides of different lengths and selected the

one ($L/\lambda=9.2$) that provided the largest gain/attenuation as the final optimized device, because there is a critical length of the waveguide at which the maximum gain can be achieved, as shown in Figure R12 below.

Figure R12 | The signal gains versus the propagation length under distinct phase differences $\varphi_p - 2\varphi_s$.

However, the operating linewidth of our structure has not been well optimized. To achieve a broad bandwidth, the dispersion curve should be well designed to keep the phase mismatch (out of the phase matching frequency) small and the optimal waveguide length short. Detailed discussions on the linewidth of our system are given on pages 4-6 of Supplementary Information.

Comment:

2. It seems from Fig. 2 that the amplification or attenuation is applied to the signal wave, instead of the pump wave. I wonder if it is possible to apply to the pump wave instead. In other words, what happens if the signal wave has a much larger amplitude than the pump wave? Or in general, is it possible to control a wave of larger amplitude via a wave of smaller amplitude, like a triode.

Our Reply:

Thank you for your question. If the signal wave has a much larger amplitude than the pump wave, we can use the signal wave to control the pump wave. In this case, it is equivalent to exchanging the frequency of the signal wave and the pump wave in the original case, i.e., $\omega_s = 2\omega_p$ and $k'_s = 2k'_p$. Consider the co-propagating signal and pump waves with the electric fields given by $E_s(z) = A_s(z)e^{i(k'_s + ik''_s)z - 2i\omega_p t}$ and $E_p(z) = A_p(z)e^{i(k'_p + ik''_p)z - i\omega_p t}$. When $A_p(z) \gg A_s(z)$, we can assume that $A_p(z)$ is undepleted and $\frac{dA_p(z)}{dz} \approx 0$, so $A_p(z)$ can be written as A_p . In this case, the nonlinear coupled mode equation can be written as

$$\frac{dA_s(z)}{dz} = \frac{2i\omega_p^2}{k_s c_0^2} \chi_{\text{eff}}^{(2)} A_p^2 e^{-i\Delta k z},$$

where $\Delta k = (k'_s - 2k'_p) + i(k''_s - 2k''_p) = i(k''_s - 2k''_p)$. The general solution of the equation above can be found as,

$$A_s(z) = i \frac{2\omega_p^2}{(k''_s - 2k''_p) k k_s c_0^2} \chi_{\text{eff}}^{(2)} A_p^2 e^{-i\Delta k z} + C$$

where the constant C can be determined by the initial conditions at $z = 0$:

$$A_s(0) = i \frac{2\omega_p^2 \chi_{\text{eff}}^{(2)} A_p^2}{(k''_s - 2k''_p) k k_s c_0^2} + C \Rightarrow C = A_s(0) - i \frac{2\omega_p^2 \chi_{\text{eff}}^{(2)} A_p^2}{(k''_s - 2k''_p) k k_s c_0^2}.$$

Now we have got

$$E_s(z) = A_s(z) e^{ik'_s - k''_s z - 2i\omega_p t}.$$

The signal gain can be calculated as,

$$G = \left| \frac{E_s(L)}{E_s(0)} \right|^2 = \left| 1 + i \frac{2\omega_p^2 \chi_{\text{eff}}^{(2)} A_p^2}{(2k''_p - k''_s) k k_s c_0^2 A_s(0)} (1 - e^{-i\Delta k L}) \right|^2 e^{-2k''_s L}$$

Setting the initial phases of the signal wave and pump wave as φ_s and φ_p , then

$A_s(0)$ and constant A_p can be expressed as $|A_s|e^{i\varphi_s}$ and $|A_p|e^{i\varphi_p}$, so the signal gain

can be written as

$$G = \left| 1 + i \frac{2\omega_p^2 \chi_{\text{eff}}^{(2)} |A_p|^2}{(2k_p'' - k_s'') k_s c_0^2 |A_s|} e^{i(2\varphi_p - \varphi_s)} (1 - e^{-i\Delta k L}) \right|^2 e^{-2k_s'' L}$$

Comparison of the equation above with Eq. (3) in the manuscript shows that the biggest difference from the case of $\omega_p = 2\omega_s$ is that, when $\omega_s = 2\omega_p$, in addition to $|A_p|$, $|A_s|$ also has an effect on the gain. Thus, by making the input signal power small enough, a weak pump wave is also able to control the signal wave, which is not possible in $\omega_p = 2\omega_s$ case.

Figure R13 | Counterplot of the signal gain/absorption in terms of the propagation length L/λ and (a) loss tangent of the nonlinear medium or (b) signal power in $\omega_s = 2\omega_p$ case.

We plot the gain/absorption of the signal wave in terms of the propagation distance L/λ and loss tangent of the nonlinear medium or signal power in $\omega_s = 2\omega_p$ case in Figure R13. It shows that for a fixed length of nonlinear waveguide, its loss tangent must be a specific value to achieve CPA. In contrast, the $\omega_p = 2\omega_s$ case is insensitive to the loss tangent and can achieve CPA a large range of lengths. Figure R13 also illustrates that the input signal power affects the signal gain, which imply that the signal power that can be fully absorbed on different lengths of waveguide varies at the same pump power.

The equation above demonstrates that the phase difference between the input waves $2\varphi_p - \varphi_s$ will influence the signal gain. Similar to the $\omega_p = 2\omega_s$ case, the maximum attenuation is achieved at $2\varphi_p - \varphi_s = 90^\circ + \arg(k_s)$ when $\omega_s = 2\omega_p$, in

which $\arg(k_s) \approx 0.829^\circ \approx 1^\circ$. As shown in Figure R14, we measured the signal gain versus the input signal power as well as the phase difference $2\varphi_p - \varphi_s$ when the pump power is about -1.73 dBm and the waveguide length is 9.2λ , and the measured results are in good agreement with the calculated results. The dip in Figure R14(a) is very sharp so the CPA can only be achieved at specific signal powers. In contrast, in $\omega_p = 2\omega_s$ case signal waves at any input power can be perfectly absorbed as long as in the nondepletion pump regime. In addition, from Figure R14(b) we can see that tuning the phase difference can adjust the signal attenuation, but the modulation range is not large enough for realizing perfect transmission. In conclusion, controlling a high-frequency wave with a low-frequency wave is not as widely applicable as controlling a low-frequency wave with a high-frequency wave.

Figure R14 | The calculated and measured signal gains of the nonlinear SSPP waveguide. (a) The signal gains versus the signal power. (b) The signal gains versus the phase difference between the signal and pump waves.

Both the $\omega_p = 2\omega_s$ case and the $\omega_s = 2\omega_p$ case need to use a wave of much larger amplitude to control a wave of small amplitude. It is impossible to use our method to control a wave of larger amplitude via a wave of smaller amplitude.

The above discussions can be found on Pages 9-12 of the revised supplementary information.

Comment:

3. What happens if the signal and pump waves have similar amplitudes. Usually what is the criteria on the ratio between them?

Our Reply:

Thank you very much for your good question. If the signal and pump powers have similar amplitudes, the amplification/absorption effect will be very poor. For the PA case, because the pump wave power is transferred to the signal wave through the nonlinear process, the pump power will drop drastically if the initial pump and signal waves have similar power amplitudes. Then the signal only has weak amplification due to the small pump power (As we have discussed above, the pump power has critical influence on the gain). In the CPA case, if the pump power is not high enough (e.g. comparable to the signal power), the generated differential wave which is out of phase with the signal wave will be very weak, so only a small portion of the signal wave will be absorbed.

In order to find the power ratio between the signal wave and the pump wave that gives our SSPP device the maximum modulation depth, we measured the gains at different signal input powers when the pump power is fixed at 21.59dBm. Note that as the signal power increases, the undepletion approximation fails, and the signal gain will fall. Figure R15 indicates that the higher gain requires lower signal power, and that the power ratio of the pump wave to the signal wave should be greater than $21.59\text{dBm}/-25\text{dBm} \approx 45604$ to achieve largest modulation depth of the device. These discussions have been added on Pages 7-8 of the revised supplementary information.

Figure R15 | The signal gains versus the signal power.

Comment:

Will the structure of effective nonlinear media induce reflection? I understand that here the input port may reflect the reflected wave back into the structure again, but in this way, the length between the port and the structure might be important and worth mentioning. Since the absorption is discussed, it is better to make a clearer clarification.

Our Reply:

Thank you very much for your good question. We have designed the reference impedance at the ports to be about 50Ω to match the port impedance of the coaxial lines, so no back and forth reflections occur. The S-parameter curves for our SSPP structure are shown in Figure R16. The measured reflection coefficient of the SSPP waveguide is smaller than -10 dB around the phase mating point 4.46 GHz (meaning that the reflected energy is less than 10%), indicating that our case can be approximated as non-reflective, and the maximum absorption rate (when the measured gain is -33dB) is higher than 90%.

Figure R16 | The simulated and measured S-parameter curves of the SSPP waveguide.

Comment:

5. I notice that the authors may have missed some related important papers of CPA in the reference list, e.g. Physical Review A 94, 063841 (2016), EPL 114, 28003 (2016), Laser & Photonics Reviews 12, 1800001 (2018), Applied Physics Letters 120, 171703 (2022), Science 377, 995 (2022). I suggest the authors add them and discuss appropriately.

Our Reply:

Thank you very much for your kind reminder. We have added these important papers of CPA into the reference list and the related discussions in introduction section of the revised manuscript.

REVIEWERS' COMMENTS

Reviewer #1 (Remarks to the Author):

The authors have replied to my comments, and some changes related to the comments appear in the new version. As a result the manuscript has substantially improved. I recommend to publish in Nature Communications.

Reviewer #3 (Remarks to the Author):

The authors have addressed all my comments well and substantially modified the manuscript. I think this is an interesting and important work and thus recommend publication on Nature Communications.